# Identification and Combinatorial Overexpression of Key Genes for Enhancing ε-Poly-L-lysine Biosynthesis in *Streptomyces albulus*

**Hongjian Zhang †, Hao Yang †, Chongyang Zhang, Daojun Zhu, Liang Wang, Jianhua Zhang and Xusheng Chen \***

Key Laboratory of Industrial Biotechnology, Ministry of Education, School of Biotechnology, Jiangnan University, Wuxi 214122, China; hjzhang@jiangnan.edu.cn (H.Z.); 18752112690@163.com (H.Y.); zhangchongyang@cathaybiotech.com (C.Z.); zhu_daojun@163.com (D.Z.); wangl@jiangnan.edu.cn (L.W.); jhzh882@163.com (J.Z.)

\* Correspondence: chenxs@jiangnan.edu.cn

† These authors contributed equally to this work.

**Abstract:** ε-Poly-L-lysine (ε-PL) is a natural and safe food preservative mainly produced by the aerobic, filamentous bacterium *Streptomyces albulus*. Therefore, it is crucial to breed superior ε-PL-producing strains to enhance fermentation efficiency to reduce production costs. Metabolic engineering is an effective measure for strain modification, but there are few reports on key genes for ε-PL biosynthesis. In this study, metabolic flux analysis was employed to identify potential key genes in ε-PL biosynthesis in *S. albulus* WG-608. A total of six potential key genes were identified. Three effective key genes (*ppc*, *pyc* and *pls*) were identified for the first time in ε-PL biosynthesis through overexpression experiments. It also presents the first demonstration of the promoting effects of *ppc* and *pyc* on ε-PL biosynthesis. Three genes were then co-expressed in *S. albulus* WG-608 to obtain OE-*ppc-pyc-pls*, which exhibited an 11.4% increase in ε-PL production compared to *S. albulus* WG-608, with a 25.5% increase in specific ε-PL production. Finally, the metabolic flux analysis of OE-*ppc-pyc-pls* compared to *S. albulus* WG-608 demonstrated that OE-*ppc-pyc-pls* successfully altered the metabolic flux as expected. This study not only provides a theoretical basis for the metabolic engineering of ε-PL-producing strains but also provides an effective approach for the metabolic engineering of other metabolites.

**Keywords:** ε-poly-L-lysine; *Streptomyces albulus*; metabolic flux analysis; metabolic engineering modification; fed-batch fermentation

## 1. Introduction

ε-Poly-L-lysine (ε-PL), one of eight natural homopoly(amino acid)s [1], is a biomolecule consisting of 25~35 L-lysine residues linked by ε-amino and α-carboxylic groups, with molecular weights ranging from 3200 to 4500 Da [2]. ε-PL has excellent broad-spectrum antibacterial activity, including Gram-positive and Gram-negative bacteria, yeasts, and fungi, mainly due to its heterotypic peptide bond and polyamino structure [3]. ε-PL has been approved as a Generally Recognized as Safe (GRAS) product by the US Food and Drug Administration and authorized as a natural food preservative in Japan, Korea, the United States, and China [4]. Moreover, due to its commercially valuable properties such as environmental and human safety and biodegradability, ε-PL is extensively utilized as a therapeutic agent, disruptor, enzyme protector, drug carrier, etc. [5].

Microbial fermentation is the main method for industrial-scale production of ε-PL due to its practicality, efficiency, cost-effectiveness, and environmental sustainability. Many microorganisms in nature, such as *Streptomyces* [6], *Kitasatospora* [7], *Epichloë* [8], *Bacillus* [9], and *Corynebacterium* [10], have been reported to produce ε-PL. *Streptomyces albulus* [11] is a widely known ε-PL-producing strain. However, the ε-PL production of these wild-type strains is extremely low (usually less than 0.2 g/L), which poses a challenge for

meeting the requirements of industrial production. In recent decades, physicochemical mutagenesis, mutagenesis combined with drug resistance screening, protoplast fusion, and ribosome engineering have been widely employed to enhance the ability of these producing strains to synthesize ε-PL [5]. As shown in Table 1, Xiang, Wang, Hiraki, Li, and Zhou et al.'s [12–16] treatment of *S. albulus* with various mutagenesis techniques enhanced ε-PL production by 0.6–2.8-fold. However, traditional mutagenesis suffers from randomness, high workload, and long lead times. Fortunately, with the advancement of metabolic engineering, new strategies have been provided for the continued engineering of ε-PL-producing strains (Table 1). For example, the synthesis of ε-PL could be enhanced through metabolic engineering by enabling the removal of the feedback inhibition of L-lysine [17] to alleviate oxygen limitation [18] and to block the synthesis of by-products [19]. However, Wang et al. [20] overexpressed *metK* in *S. albulus* NK660, and no significant increase in ε-PL production was observed, indicating that finding the key metabolic nodes and genes that limit ε-PL synthesis is essential for modifying ε-PL-producing strains using metabolic engineering.

**Table 1.** Overview of the production of ε-PL by engineered *S. albulus* in fed-batch fermentation.

| Strains | Breeding Strategies | Fermentation Strategies | Production (g/L) | Reference |
|---|---|---|---|---|
| **Traditional Mutagenesis** | | | | |
| *S. albulus* 11011A | AEC, Gly resistance screening | Constant pH 4.0 | 20.0 | [14] |
| *S. albulus* S410 | AEC, Gly resistance screening | Two-stage pH control | 48.3 | [21] |
| *S. albulus* SAR14 | ARTP mutagenesis | Shake flask fermentation | 1.1 | [12] |
| *S. albulus* UN2-71 | Nitrite, UV, AEC compound treatment | Shake flask fermentation | 1.6 | [22] |
| *S. albulus* F3-4 | Genome shuffling | Constant pH 3.8 | 13.5 | [15] |
| *S. albulus* FEEL-1 | Ribosome engineering | Constant pH 4.0 | 24.5 | [13] |
| *S. albulus* F4-22 | Genome shuffling | Constant pH 4.0 | 40.0 | [16] |
| *S. albulus* AG3-28 | Gentamicin screening | Constant pH 3.8 | 56.5 | [23] |
| *S. albulus* M-Z18 | UV and NTG mutagenesis | pH shock | 32.2 | [24] |
| *S. albulus* R6 | ARTP mutagenesis, antibiotic screening | pH shock | 70.3 | [25] |
| *S. albulus* GS114 | Streptomycin resistance screening | Dynamic pH control | 60.2 | [26] |
| **Metabolic engineering modifications** | | | | |
| *S. albulus* CR1-*ask* | *Ask* gene targeted mutation | Constant pH 4.0 | 15.0 | [17] |
| *S. albulus* PD-2 | Overexpression of the *vgb* | Two-stage pH control | 34.2 | [18] |
| *S. albulus* PD-1 | Overexpression of the *amtB* | Two-stage pH control | 35.7 | [27] |
| *S. albulus* PD-5 | Knockout of *plsI* overexpression of *plsII* | Two-stage pH control | 23.6 | [28] |
| *S. albulus* NBRC14147 | Overexpression of *ttm* and *nys* | Two-stage pH control | 3.5 | [29] |
| *S. albulus* Q-PL2 | Overexpression of *pls* | Constant pH 4.0 | 20.1 | [4] |
| *S. albulus* PL05 | Overexpression of *pap* and *ppk2B^cg* | Constant pH 4.0 | 59.25 | [30] |

Metabolic flux analysis is one of the pillars of metabolic engineering. It is an effective means for identifying key metabolic nodes. Over the past decades, it has been widely used to quantify intracellular metabolic fluxes [31]. Refining the metabolic flux distribution of different conditions or pathways can characterize the metabolic capacity of cells [32], unveil the impact of genetic modifications on cell metabolism, and establish a theoretical foundation for subsequent metabolic engineering modifications [33]. Cheah et al. [34] applied isotopically nonstationary metabolic flux analysis to recombinant aldehyde-producing cyanobacteria and found that the fluxes of pyruvate dehydrogenase (PDH) and phosphoenolpyruvate carboxylase (PPC) were inversely correlated with product formation, and PDH knockdown led to a 47% increase in aldehyde productivity. To improve *Basfia succiniciproducens* succinate yield, Becker et al. [35] utilized systems-wide $^{13}$C metabolic flux analysis to determine the undesired fluxes through pyruvate-formate lyase (PflD) and lactate dehydrogenase (LdhA) and subsequently knocked down the PflD as well as LdhA, revealing a 45% increase in succinate yield in the double deletion strain *B. succiniciproducens* ΔldhA ΔpflD. The above studies demonstrate that using metabolic flux analysis for identifying the key genes in ε-PL biosynthesis in the ε-PL-producing strain is feasible.

In this study, we performed metabolic flux analysis of ε-PL high- and low-producing strains to understand the key metabolic nodes and regulatory mechanisms of ε-PL biosynthesis in *S. albulus*. Then, we mined six key genes that may affect ε-PL biosynthesis. Subsequently, the effect of each candidate key gene on ε-PL biosynthesis was evaluated by

gene overexpression. Three genes, *ppc* (encoding phosphoenolpyruvate carboxylase), *pyc* (encoding pyruvate carboxylase), and *pls* (encoding ε-PL synthetase), were identified for the first time in ε-PL biosynthesis through overexpression experiments. Finally, *ppc*, *pyc*, and *pls* were co-expressed in *S. albulus* WG-608 to obtain OE-*ppc-pyc-pls* that successfully increased ε-PL production and specific ε-PL production, and metabolic flux was successfully altered as expected.

## 2. Materials and Methods

### 2.1. Strains, Plasmids, and Media

The strains and plasmids utilized in this study are itemized in Table 2. *S. albulus* M-Z18 [24] is maintained by the China Center for Type Culture Collection (CCTCC M2019589), and *S. albulus* WG-608 [36] was used as the original strain for metabolic engineering modification in this study. *E. coli* DH5α was used as a cloning and expression host. *E. coli* ET12567/pUZ8002 was used for *Streptomyces–E. coli* interspecies conjugation to facilitate plasmid introduction into *Streptomyces*.

**Table 2.** Strains and plasmids used in this study.

| Strain or Plasmid | Description | Source or Reference |
|---|---|---|
| | **Strains** | |
| | *S.albulus* | |
| M-Z18 | Parent strain, ε-poly-L-lysine low-production strain | [24] |
| WG-608 | Parent strain, ε-poly-L-lysine high-production strain | [36] |
| OE-*ppc* | WG608 carrying pIB139-*ppc* | This study |
| OE-*zwf* | WG608 carrying pIB139-*zwf* | This study |
| OE-*dapA* | WG608 carrying pIB139-*dapA* | This study |
| OE-*lysA* | WG608 carrying pIB139-*lysA* | This study |
| OE-*pyc* | WG608 carrying pIB139-*pyc* | This study |
| OE-*pls* | WG608 carrying pIB139-*pls* | This study |
| Control-pIB139 | WG608 carrying pIB139 | This study |
| OE-*ppc-pyc-pls* | WG608 carrying pIB139-*ppc-pyc-pls* | This study |
| | *E. coli* | |
| DH5α | General cloning host | Invitrogen |
| ET12567 | Donor strain for conjugation between *E. coli* and *Streptomyces* | Invitrogen |
| | **Plasmids** | |
| pIB39 | Integrative vector based on φC31 integrase | [37] |
| pIB139-*ppc* | *ppc* cloned in pIB139 | This study |
| pIB139-*zwf* | *zwf* cloned in pIB139 | This study |
| pIB139-*dapA* | *dapA* cloned in pIB139 | This study |
| pIB139-*lysA* | *lysA* cloned in pIB139 | This study |
| pIB139-*pyc* | *pyc* cloned in pIB139 | This study |
| pIB139-*pls* | *pls* cloned in pIB139 | This study |
| pIB139-*ppc-pyc-pls* | *ppc*, *pls* and *pyc* cloned in pIB139 | This study |

The *E. coli* strains were cultured in Luria–Bertani medium (10 g/L NaCl, 10 g/L tryptone, 5 g/L yeast extract, pH 7.0) supplemented with final concentrations of chloramphenicol (25 μg/mL), apramycin (50 μg/mL), and kanamycin (25 μg/mL) as needed at a temperature of 37 °C. Spores of *S. albulus* WG-608 and its derivatives were produced on BTN medium (1 g/L yeast extract powder, 2 g/L fish meal peptone, 10 g/L glucose, 20 g/L agar powder, pH 7.5) at a temperature of 30 °C [25]. MS medium containing 10 mM $MgCl_2$, 20 g/L soybean powder, 20 g/L mannitol, and 20 g/L agar powder (pH 7.0) was used for intergeneric conjugation between *S. albulus* and *E. coli* at 30 °C. Furthermore, the final concentrations of 50 μg/mL apramycin and 25 μg/mL nalidixic acid were superimposed on MS medium to screen for recombinant colonies. YP medium containing 10 g/L yeast powder, 60 g/L glucose, 0.2 g/L $K_2HPO_4 \cdot 3H_2O$, 0.04 g/L $ZnSO_4 \cdot 7H_2O$, 0.5 g/L $MgSO_4 \cdot 7H_2O$, 0.03 g/L $FeSO_4 \cdot 7H_2O$, 5 g/L $(NH_4)_2SO_4$ (pH 6.8) was used for shake-flask fermentation of *S. albulus* derivatives [25]. YHP medium (10 g/L yeast powder, 60 g/L glucose, 4 g/L $KH_2PO_4$, 5 g/L $(NH_4)_2SO_4$, 0.04 g/L $ZnSO_4 \cdot 7H_2O$, 0.5 g/L $MgSO_4 \cdot 7H_2O$, 0.03 g/L $FeSO_4 \cdot 7H_2O$, pH 6.8) was utilized for the cultivation of overexpression strains in a 5-L fermenter [25].

### 2.2. Plasmid and Strain Construction

All strains and plasmids used in this study are listed in Table 2, and all primers are listed in Supplementary Materials: Table S1. To achieve overexpression of *pyc*, *ppc*, *zwf*, *dapA*, *lysA*, and *pls* in *S. albulus* WG-608, the method was performed as described by Hu et al. [38] with some modifications. For overexpression of *pyc*, a DNA fragment containing the *pyc* coding region was amplified from genomic DNA from *S. albulus* WG6-08 using primer pair *pyc*-F/-R. The fragment was purified and ligated into the vector pIB139 [37], which was digested *Nde* I and *Eco*RI. Subsequently, ligation product was then transformed into competent E. coli DH5α, and exconjugants were picked out from LB plates containing 50 μg/mL apramycin. After validation by colony PCR using corresponding primers and DNA sequencing (Aenta, Suzhou, China), the overexpression vectors pIB139-*pyc* were obtained. The DNA fragments encoding *ppc*, *zwf*, *dapA*, *lysA*, and *pls* were chemically synthesized (Aenta, Suzhou, China) with codon optimization, and other genes were amplified as described above and recombinant plasmids were constructed.

To achieve co-expression of multiple genes in *S. albulus* WG-608, the method was performed as described by Rao et al. [39] with some modifications. The DNA fragment containing *pyc* was amplified from pIB139-*pyc*, the DNA fragment containing *ppc* was amplified from pIB139-*ppc*, and the DNA fragment containing *pls* was amplified from pIB139-*pls*. These three fragments were ligated into the vector pIB139 digested with *Nde*I to obtain the co-expression vector pIB139-*ppc-pyc-pls*.

Finally, the above recombinant plasmids were separately transformed into *E. coli* ET12567 for intergeneric conjugation with *S. albulus* WG-608. Transformants were screened on BTN solid medium supplemented with apramycin and nalidixic acid, and the colonies were verified by PCR using the corresponding primers.

### 2.3. Shake-Flask, Batch, and Fed-Batch Fermentation of S. albulus

For shake-flask fermentation, *S. albulus* WG-608 and its mutants were cultured in YP medium, at 30 °C, 200 rpm for 24 h as seed culture. Then, the seeds were inoculated with an inoculation amount of 8% and continued to be cultured in YP medium for 72 h, at 30 °C, 200 rpm.

Batch fermentation was performed as described by Pan et al. [24] with some modifications. The batch fermentations of *S. albulus* WG-608 and its mutants were performed in a 1-L bioreactor system using a constant pH strategy. The YHP medium was agitated at 300 rpm with a mechanical stirrer (Qiangle, Suzhou, China) at 30 °C with a 1 vvm aeration rate. We adjusted the initial pH to 6.80 by adding ammonia solution, and then 60 mL (inoculation amount of 8%) of 24 h old seed was inoculated into the YHP medium. The pH level of the fermentation broth was maintained at 4.0 by automatic addition of ammonia solution when it spontaneously dropped to 4.0. The DO level was maintained above 20% of air saturation controlled by adjusting agitation speed from 200 to 1000 rpm until the end of fermentation. Fermentation ended when the glucose in the medium was depleted.

In the fed-batch fermentation, the method was performed as described by Pan et al. [24] with some modifications. Fermentation in a 5 L glass stirred tank bioreactor (Biotech-5BG, Baoxing Bio-Engineering Equipment Co., Ltd., Shanghai, China) was performed by filling in 3.5 L YHP medium. At the early fermentation stage, the conditions were the same as that in the 1 L batch fermentation. However, the agitation speed varied from 200 to 900 rpm, and the aeration rate was manually increased stepwise with steps of 0.5 vvm and a range of 0.5–2.5 vvm. In addition, when the glucose concentration in the fermentation was below 10 g/L, sterile glucose (75%, *w/v*) was automatically added and maintained at about 10 g/L to prevent carbon limitation. When the ammonia nitrogen ($NH_4^+$-N) concentration decreased below 0.5 g/L, sterilized $(NH_4)_2SO_4$ solution (40%, *w/v*) was automatically added and maintained at about 0.5 g/L to prevent nitrogen limitation.

### 2.4. Fermentation Parameters Analysis

Fermentation parameters were determined as described by Yang et al. [30] with some modifications. Determination of dry cell weight (DCW): 10 mL aliquots of culture broth were centrifuged at 12,000 rpm for 10 min and the precipitate was collected. The mycelia were filtered through a pre-weighed filter paper and dried at 105 °C to a constant weight before measuring the biomass. The supernatant was used to determine the ε-PL concentration according to the procedure described by Itzhaki [40]. Glucose concentration was determined by a biosensor (SBA-40B, Shandong Academy of Sciences, Jinan, China) through the enzymatic reaction of glucose oxidase. $NH_4^+$-N was analyzed through a colorimetric method using Nessler reagent [41].

### 2.5. RNA Sample Preparation and RT–PCR and RT–qPCR Analysis

RNA was extracted according to the instructions of Total RNA Extractor (Trizol) (Sangon, Shanghai, China). Its purity and mass concentration were determined using a nucleic acid quantifier, and RNA integrity was detected using nucleic acid electrophoresis. The isolated RNA was reverse transcribed after digestion of residual gDNA using HiScript® III RT SuperMix for qPCR (+gDNA wiper) (Vazyme, Nanjing, China). For RT-PCR the transcript levels of genes were determined in triplicate by RT-qPCR (primer pairs listed in Supplementary Materials: Table S1) for each transcript using ChamQ Universal SYBR qPCR Master Mix (Vazyme, Nanjing, China). The housekeeping gene *hrdB* was utilized as an internal control to normalize samples and quantified using the $2^{-\Delta\Delta CT}$ method [42].

### 2.6. Metabolic Network of ε-PL Synthesis and Construction of An Econometric Model

Metabolic flux analysis is based on a metabolic network model, and the metabolic fluxes of the ε-PL synthesis process need to be analyzed in conjunction with the metabolic pathways involved in the ε-PL synthesis process. The metabolic networks involved in cellular metabolism are complex and must be refined. Whole genome sequencing results showed that *S. albulus* has a complete pentose phosphate pathway, EMP pathway, TCA pathway, back-complementation pathway, DAP pathway, and ε-PL synthesis pathway, so all these major carbon metabolic pathways were included in the metabolic network map of ε-PL synthesis [43] Due to the lack of data on the cellular composition of *S. albulus*, differences of up to 20% in cellular components would not affect the distribution of metabolic fluxes in *S. lividans* [44]. Since *Bacillus subtilis* is capable of synthesizing ε-polyglutamic acid, which is a non-ribosomal enzyme synthesis system like the synthesis of ε-PL by *S. lividans* [45], *B. subtilis* was chosen as a reference, and the bacterial molecular formula ($CH_{1.724}O_{0.524}N_{0.2}$) [46] was used to construct the metabolic network of *S. albulus* M-Z18 and *S. albulus* WG-608 (Supplementary Materials: Figure S1).

Assuming that all intracellular intermediate metabolites are in the metabolic steady-state [46], i.e., their concentration changes at a rate of 0. Then, we have

$$d_{Xmet}/dt = O, \tag{1}$$

Xmet is the concentration vector of intracellular intermediate metabolites. In matrix form, it is expressed as:

$$S \cdot r = O, \tag{2}$$

S is the measurement matrix of this network a × b, a represents the number of intracellular reactions, b represents the number of intermediate metabolites, and r represents the metabolic reaction rate vector. The following chemical reaction rate equation is shown in Supplementary Materials: Table S2.

The reaction rate equations of the metabolic nodes are listed for each component of the *S. albulus* ε-PL metabolic network according to the law of mass conservation, as shown in Supplementary Materials: Table S3.

Based on the metabolic reaction rate equations and the number of metabolites, 23 metabolic equations and 43 unknowns were established. It has been shown that 20 common amino acids

play an important role in cytoplasmic synthesis. Therefore, the analysis of metabolic flow distribution focused on the metabolism of amino acids [47]. Changes in the concentration of 17 amino acids, ε-PL concentration, glucose consumption, and cell growth between sampling intervals were determined at the mid-log growth stage of the replenishment batch fermentation. This resulted in 20 reaction rate values. The metabolic flux for glucose consumption was defined as 100, and the remaining 19 reaction rate values were converted to their corresponding metabolic fluxes. Finally, the remaining 17 unknown metabolic fluxes were solved.

### 2.7. Statistical Analysis

All experiments were conducted in triplicate, and all data were expressed as mean ± standard deviation. The SPSS (version 22.0, SPSS Inc., Chicago, IL, USA) was used for statistical analysis that was performed using one-way analysis of variance (ANOVA) and Tukey's test at $p < 0.05$.

## 3. Results

### 3.1. Fermentation Profiles of S. albulus WG-608 and S. albulus M-Z18

To identify key nodes in ε-PL synthesis in *S. albulus* WG-608, metabolic flux analysis of the ε-PL synthesis pathway was employed to pinpoint crucial information for subsequent metabolic engineering modifications. As shown in Figure 1a, *S. albulus* WG-608 was obtained by iterative mutagenesis and selection using *S. albulus* M-Z18 as the original strain. We found that the cell growth of *S. albulus* WG-608 was significantly weaker than *S. albulus* M-Z18. In contrast, the ε-PL production of *S. albulus* WG-608 was 84.8% higher than *S. albulus* M-Z18 (Figure 1b), indicating that *S. albulus* WG-608 had a significantly greater capacity to synthesise ε-PL compared to *S. albulus* M-Z18. This implies significant differences in ε-PL synthesis between *S. albulus* WG-608 and *S. albulus* M-Z18, rendering it plausible to scrutinize the potential nodes involved in this process via metabolic flux analysis.

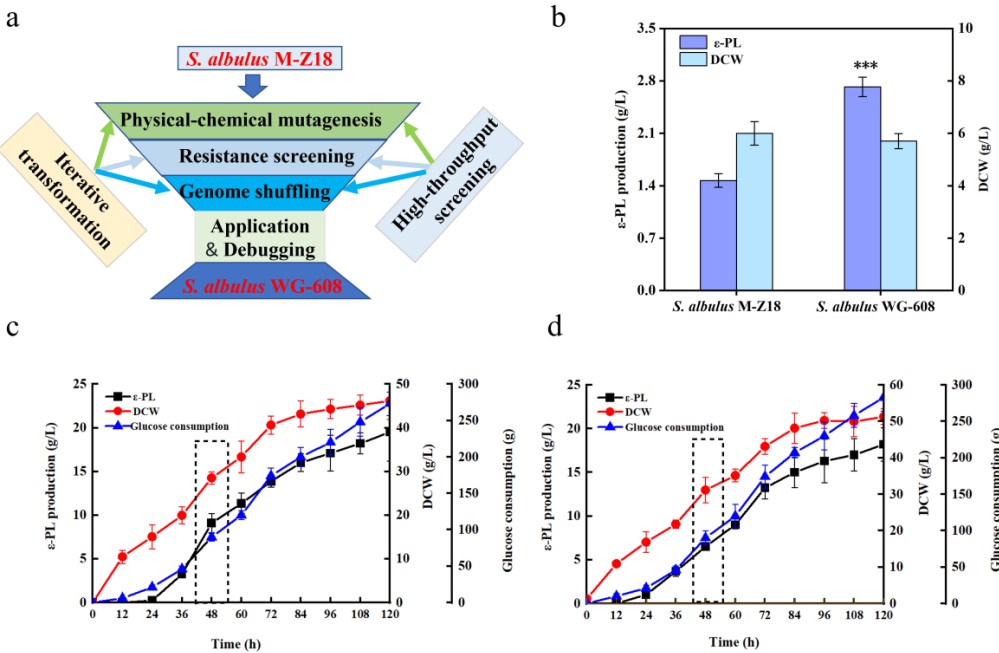

**Figure 1.** Comparison of *S. albulus* WG-608 and *S. albulus* M-Z18 and its fed-batch fermentation. (**a**) Multiple mutagenesis iterative breeding method; (**b**) comparison of shake flask fermentation of *S. albulus* WG-608 and *S. albulus* M-Z18; (**c**) fed-batch fermentation of *S. albulus* WG-608; (**d**) fed-batch fermentation of *S. albulus* M-Z18. The data are presented as averages, and the error bars represent standard deviations ($n = 3$). *** $p < 0.001$.

The growth status of the strain at the metabolic steady-state serves as the basis for metabolic flux analysis. Cell growth, ε-PL synthesis, and glucose consumption peaked dur-

ing mid-log growth at 42–54 h (Figure 1c,d). Research has demonstrated a close correlation between the growth of the strain and ε-PL synthesis [48], suggesting that the strain reached a metabolic steady-state during the 42–54 h period. Therefore, we collected cell samples of *S. albulus* WG-608 and *S. albulus* M-Z18 cultivated for 42–54 h for metabolic flux analysis.

### 3.2. Comparison of the Metabolic Flux of S. albulus WG-608 and S. albulus M-Z18

The metabolic flux rate of glucose consumption (*S. albulus* M-Z18: 0.84 mmol/g·h$^{-1}$ and *S. albulus* WG-608: 0.73 mmol/g·h$^{-1}$) was defined as 100. Concentrations of 17 amino acids, ε-PL synthesis, and changes in cell growth were analyzed to determine metabolites of metabolic flux (Supplementary Materials: Table S4). The differences between *S. albulus* WG-608 and *S. albulus* M-Z18 during ε-PL synthesis were analyzed according to the final metabolic flux (Figure 2) to identify the key metabolic node in *S. albulus* WG-608 that contributes to increased ε-PL production.

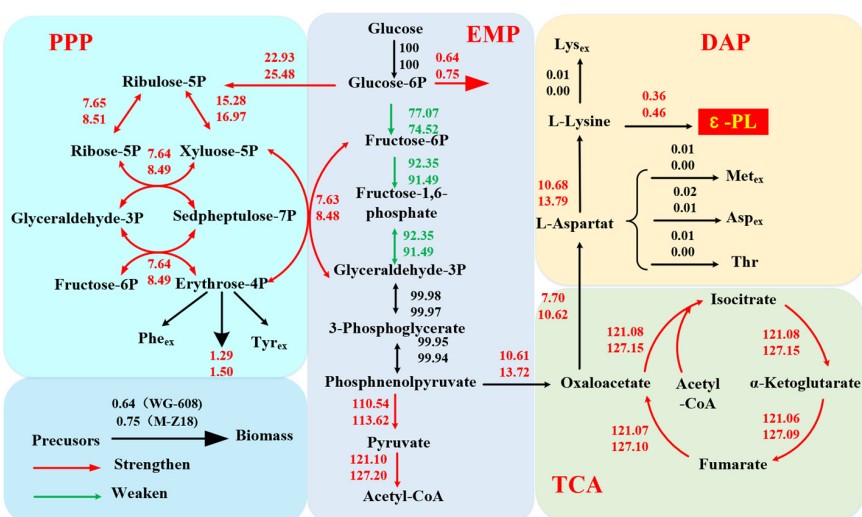

**Figure 2.** Metabolic flux distribution of *S. albulus* WG-608 and *S. albulus* M-Z18. Red arrows show the increased metabolic flux of *S. albulus* WG-608, the green arrow represents the decreased metabolic flux of *S. albulus* WG-608, the number above represents *S. albulus* WG-608, and the number below represents *S. albulus* M-Z18. Red color indicates that the metabolic flux of WG-608 is stronger than M-Z18; Green color indicates that the metabolic flux of M-Z18 is stronger thanWG-608; Black indicates that there is no differences between the metabolic flux of WG608 and M-Z18.

The rate of product synthesis is regulated by various intermediate metabolites and their corresponding enzymes, while the concerted action of key enzymes across all levels of intermediate metabolic reactions determines the production of the final product. The production of the target product is intricately linked to the allocation of carbon flux among metabolites, with all nodes in the metabolic network playing a role and exerting varying degrees of influence on the allocation of carbon flux allocation. Within the metabolic network governing ε-PL synthesis, key nodes affecting this process include glucose-6-phosphate, phosphoenolpyruvate (PEP), pyruvate, oxaloacetate (OAA), aspartate, and α-ketoglutarate. The carbon flux allocation of these nodes was analyzed to compare the flux distribution between *S. albulus* WG-608 and *S. albulus* M-Z18 to identify potential nodes for metabolic engineering to further enhance ε-PL production.

Based on the above distribution of metabolic flux of high- and low-producing strains, it is easy to find that the flux of ε-PL synthesis in *S. albulus* WG-608 was increased by 27.8% compared to *S. albulus* M-Z18. The fluxes of pentose phosphate, anaplerotic reactions, TCA, and DAP pathways were increased by 11.1%, 29.3%, 5.0%, and 16.7%, respectively. The increased metabolic fluxes of the pentose phosphate and TCA cycle pathways could provide sufficient NADPH and ATP for ε-PL synthesis. In contrast, the increased fluxes of

anaplerotic reactions and the DAP pathway could provide more carbon flux and precursor L-lysine for ε-PL biosynthesis and promote the synthesis of ε-PL.

### 3.2.1. Metabolic Flux Analysis of Glucose-6-phosphate Node

The pentose phosphate pathway serves as the primary source of intracellular NADPH in *S. albulus*, facilitated by glucose 6-phosphate dehydrogenase (encoding by *zwf*) and 6-phosphogluconate dehydrogenase (encoding by *gntZ*), which also supplies various structured sugar molecules for cell growth. Carbon flux from glucose in *S. albulus* is directed to both the pentose phosphate pathway and EMP pathway via glucose-6-phosphate. The flux from glucose-6-phosphate to the pentose phosphate pathway in *S. albulus* WG-608 was observed to increase by approximately 11.1% compared to *S. albulus* M-Z18 (Figure 3a), indicating that there is more carbon flux in the pentose phosphate pathway in *S. albulus* WG-608, which may lead to increased NADPH production.

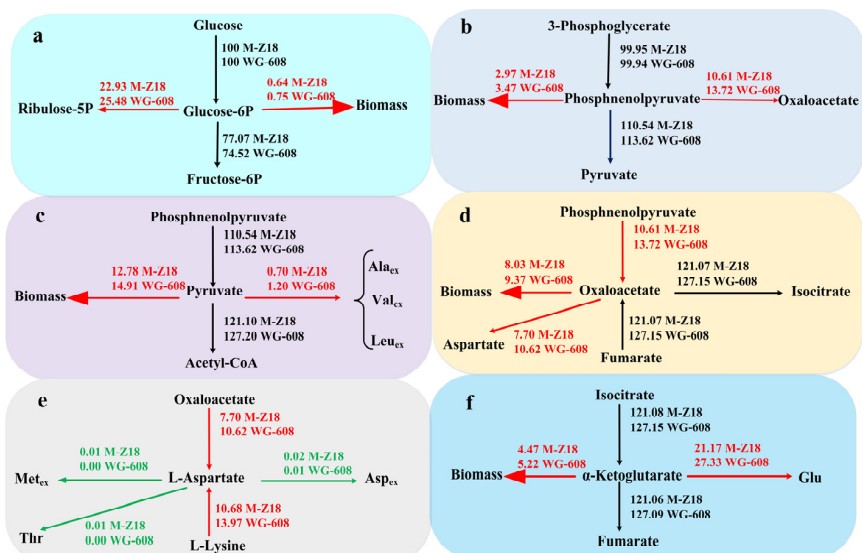

**Figure 3.** Metabolic flux distribution at key nodes. (**a**) Glucose-6-phosphate node. (**b**) Phospho-enolpyruvate node. (**c**) Pyruvate node. (**d**) Oxaloacetate node. (**e**) Aspartate node. (**f**) α-Ketoglutarate node. Red color indicates that the metabolic flux of WG-608 is stronger than M-Z18; Green color indicates that the metabolic flux of M-Z18 is stronger than WG-608; Black indicates that there is no differences between the metabolic flux of WG608 and M-Z18.

### 3.2.2. Metabolic Flux Analysis of Phosphoenolpyruvate Node

The phosphoenolpyruvate node is an important anaplerotic pathway for oxaloacetate. In bacteria, such as *Corynebacterium glutamicum*, both pyruvate carboxylase and phos-phoenolpyruvate carboxylase (PEPC) are present enzymes that can replenish OAA [49]. However, in *S. albulus*, only PEPC is available to refill the depleted oxaloacetate through phosphoenolpyruvate carboxylase (encoding by *ppc*) (PEP + $CO_2$ + ADP → OAA). As expected, the flux of oxaloacetate was found to be 29.3% higher in *S. albulus* WG-608 com-pared to *S. albulus* M-Z18 (Figure 3b). This increase in anaplerotic reaction flux has been shown to facilitate ε-PL biosynthesis and provide more precursors for L-lysine synthesis via the anaplerotic reaction [50]. These findings further support *ppc* as a promising candidate gene for engineering ε-PL production.

### 3.2.3. Metabolic Flux Analysis of Pyruvate Node

Pyruvate serves as a pivotal junction for carbon flux from the glycolytic pathway into the TCA cycle, which is a catabolic pathway in most organisms and an essential metabolic system that generates substantial amounts of free energy. As the hub of cellular metabolism, the TCA cycle produces abundant NADH and $FADH_2$, which are subsequently reoxidized

via electron transport chain and oxidative phosphorylation to produce ATP [51]. In addition, the TCA cycle intermediates provide precursors for many secondary metabolites. Thus, the transfer of carbon from the pyruvic node to the TCA cycle in *S. albulus* determines the production of ATP in the cell, as well as contributes to the synthesis of various metabolic products. In *S. albulus* WG-608, the carbon flux from pyruvate was mainly directed to alanine, valine, leucine, biomass, and TCA cycle pathways. In contrast, the flux of the three amino acids in *S. albulus* WG-608 was small but increased by 71.4% compared to *S. albulus* M-Z18, while the flux of the TCA cycle pathway was only increased by 5.04% (Figure 3c).

### 3.2.4. Metabolic Flux Analysis of Oxaloacetate Node

Oxaloacetate is a key node linking the glycolytic pathway, the TCA cycle, and the L-lysine synthesis pathway. Here, the carbon flux from phosphoenolpyruvate enters the TCA cycle to supply large amounts of NADH and $FADH_2$ for strain growth, and the other carbon flux enters the L-lysine synthesis pathway to provide the precursor L-lysine for $\varepsilon$-PL synthesis. The carbon flux of the L-lysine synthesis pathway was increased by 38.57% in *S. albulus* WG-608 compared to *S. albulus* M-Z18, while the flux of the TCA pathway via oxaloacetate was only slightly increased (Figure 3d). This suggests that the increase in $\varepsilon$-PL production in *S. albulus* WG-608 is mainly caused by the increased flux of the DAP pathway.

### 3.2.5. Metabolic Flux Analysis of Aspartate Node

Aspartate is a precursor of L-lysine synthesis and its flux distribution is directly related to the amount of L-lysine produced [52]. We found that the flux of L-lysine was increased by 29.1% in *S. albulus* WG-608 compared to *S. albulus* M-Z18, while the flux of threonine, extracellular aspartate, and extracellular methionine were all significantly lower, corresponding to the increase in L-lysine flux (Figure 3e). To further improve the availability of precursor L-lysine, we could also try to introduce more carbon flux into the L-lysine synthesis pathway with the aim of further increasing the production of $\varepsilon$-PL by reducing branched amino acids.

### 3.2.6. Metabolic Flux Analysis of $\alpha$-Ketoglutarate Node

A-Ketoglutarate is a by-product of the synthesis of L-lysine and a precursor for the synthesis of glutamate, which, in turn, provides the amino group for the synthesis of L-lysine [53]. The flux of $\alpha$-ketoglutarate to glutamate was approximately 29.1% higher in *S. albulus* WG-608 than in *S. albulus* M-Z18, corresponding to the increased $\varepsilon$-PL production (Figure 3f).

### *3.3. Screening and Validation of Key Genes of the $\varepsilon$-PL Biosynthetic Pathway*
### 3.3.1. Screening of Key Genes

According to the metabolic flux distribution of *S. albulus* WG-608 and *S. albulus* M-Z18, it can be found that the carbon flux distribution of the pentose phosphate pathway, anaplerotic reactions, and DAP in *S. albulus* WG-608 were significantly higher than *S. albulus* M-Z18, which probably affects the synthesis of $\varepsilon$-PL. However, it is unclear whether the magnitude of carbon fluxes in *S. albulus* WG-608 is appropriate for these three pathways, so we will start from them in the next study. Since the process of $\varepsilon$-PL synthesis requires the catalysis of L-lysine, ATP, and Pls [54], this study focused on the synthetic pathway of L-lysine, the biosynthetic precursor of $\varepsilon$-PL (reducing power NADPH and carbon skeleton), to maximize $\varepsilon$-PL's production efficiency. It aimed to increase the biosynthetic capacity of L-lysine while fulfilling the catalytic capacity enhancement of Pls. To this end, based on the metabolic engineering modification strategy of L-lysine-producing strains [55–59], we selected the pyruvate carboxylase gene (*pyc*), phosphoenolpyruvate carboxylase gene (*ppc*), 6-phosphoglucose dehydrogenase gene (*zwf*), dihydropyridine dicarboxylic acid synthase gene (*dapA*), diamino-heptane dioic acid decarboxylase gene (*lysA*), $\varepsilon$-PL synthase gene (*pls*), and six other target genes to evaluate their effects on $\varepsilon$-PL biosynthesis by gene

overexpression. The specific positions of several genes in the metabolic pathway are shown in Figure 4a.

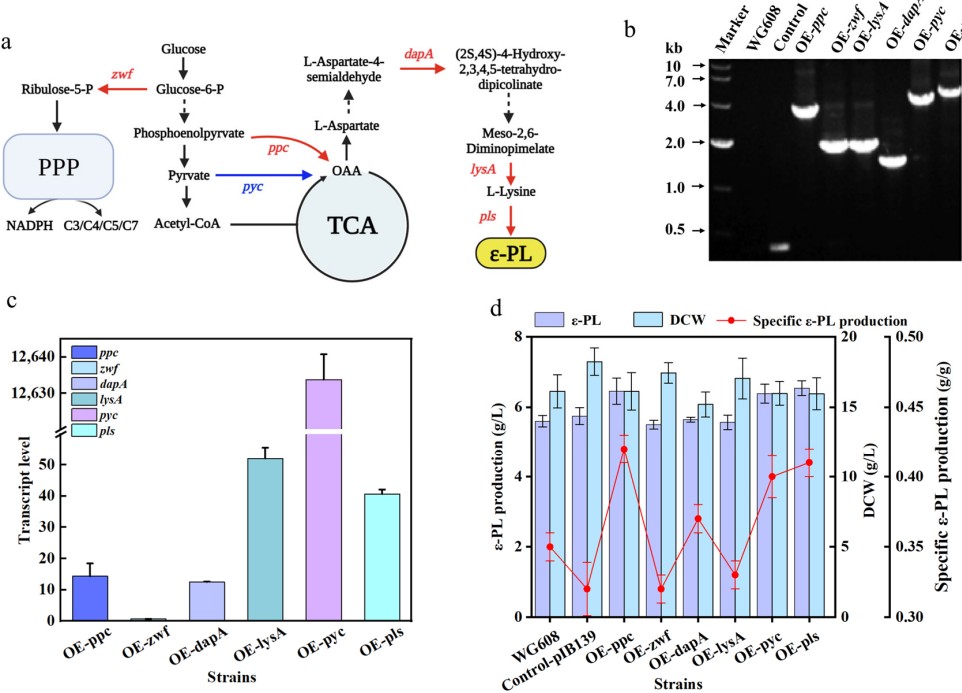

**Figure 4.** Overexpression of key genes during ε-PL synthesis and validation of batch fermentation. (**a**) The modification sites in this study: black arrows show the native metabolic network during ε-PL synthesis, red arrows show the genes overexpressed in this work, blue arrows show the exogenously expressed heterologous gene. (**b**) Genomic PCR of overexpressed strains. (**c**) RT-PCR of overexpression strains. (**d**) Batch fermentation of overexpression strains.

### 3.3.2. Construction of Key Gene Overexpression Strains

To verify whether *pyc*, *ppc*, *zwf*, *dapA*, *lysA*, and *pls* can promote the synthesis of ε-PL, the strains OE-*pyc*, OE-*ppc*, OE-*zwf*, OE-*dapA*, OE-*lysA*, and OE-*pls* were obtained by overexpression of *pyc*, *ppc*, *zwf*, *dapA*, *lysA*, and *pls* genes using pIB139 as a vector. Genomic PCR showed (Figure 4b) that all six genes were successfully integrated into the *S. albulus* WG-608 genome. RT-PCR showed Figure 4c) that *ppc* and *dapA* obtained about 15-fold expression in *S. albulus* WG-608, *lysA*, and *pls* obtained 50-fold and 40-fold expression, respectively, while *pyc* obtained 12000-fold. Surprisingly, *zwf* was not overexpressed, possibly because *S. albulus* WG-608 did not transcribe *zwf* normally or *zwf* had already obtained high expression as a housekeeping gene [60]. In conclusion, except for *zwf*, the remaining five genes, *ppc*, *dapA*, *lysA*, *pls*, and *pyc*, were successfully overexpressed in *S. albulus* WG-608.

### 3.3.3. Effect of Key Gene Overexpression on ε-PL Synthesis

Since pH is the most important environmental condition affecting ε-PL biosynthesis, the effects of *pyc*, *ppc*, *zwf*, *dapA*, *lysA*, and pls on the synthesis of ε-PL in *S. albulus* WG-608 were investigated under batch fermentation conditions at a constant pH 4.0. As shown in Figure 4d, the cell growth and ε-PL production of control pIB139 were the same as *S. albulus* WG-608, indicating that the pIB139 plasmid did not affect the cell growth and ε-PL synthesis of *S. albulus* WG-608. The ε-PL production of OE-*ppc* reached 6.5 ± 0.4 g/L, which was 15.2% higher than *S. albulus* WG-608, and the specific ε-PL production was 21.2% higher than *S. albulus* WG-608, The ε-PL production of OE-*pyc* reached 6.4 ± 0.3 g/L, which was 14.1% higher than *S. albulus* WG-608, and the specific ε-PL production was 14.3% higher than *S. albulus* WG-608. The ε-PL production of OE-*pls* reached 6.5 ± 0.2 g/L,

which was 16.8% higher than *S. albulus* WG-608, and the specific ε-PL production was increased by 17.1%. However, overexpression of *zwf*, *dapA*, and *lysA* failed to increase ε-PL production, which may be due to the limited effect of overexpression of single genes to show a difference in ε-PL production.

### 3.4. Co-Expression of Effective Key Genes Further Enhances ε-PL Synthesis

There are many examples of combinatorial overexpression of multiple genes in *Streptomyces* to enhance the target product and most of them achieve better results than single gene overexpression. Wang et al. [61] co-expressed *aroA*, *fkbN*, and *luxR* simultaneously in *Streptomyces hygroscopicus* to increase the yield of cystatin by 4.1-fold. Tao et al. [62] co-expressed *vgb*, *frr*, and *toyG* in *Streptomyces diastatochromogenes* 1628 to increase the tunicamycin production by 2.2-fold. This shows that it is feasible to further improve the ε-PL of *S. albulus* WG-608 by co-expression of effective genes.

To verify the effect of combinatorial expression of effective genes on ε-PL, we co-expressed *ppc*, *pyc*, and *pls* in *S. albulus* WG-608 to obtain OE-*ppc-pyc-pls*. Genomic PCR showed that *ppc*, *pyc*, and *pls* were successfully integrated into the genome of *S. albulus* WG-608 (Figure 5a). qRT-PCR showed that *pyc*, *pyc*, and *pls* obtained about 2-fold, 8-fold, and 3-fold expression in OE-*ppc-pyc-pls*, respectively, compared to *S. albulus* WG-608 (Figure 5b). This indicates that OE-*ppc-pyc-pls* was successfully constructed.

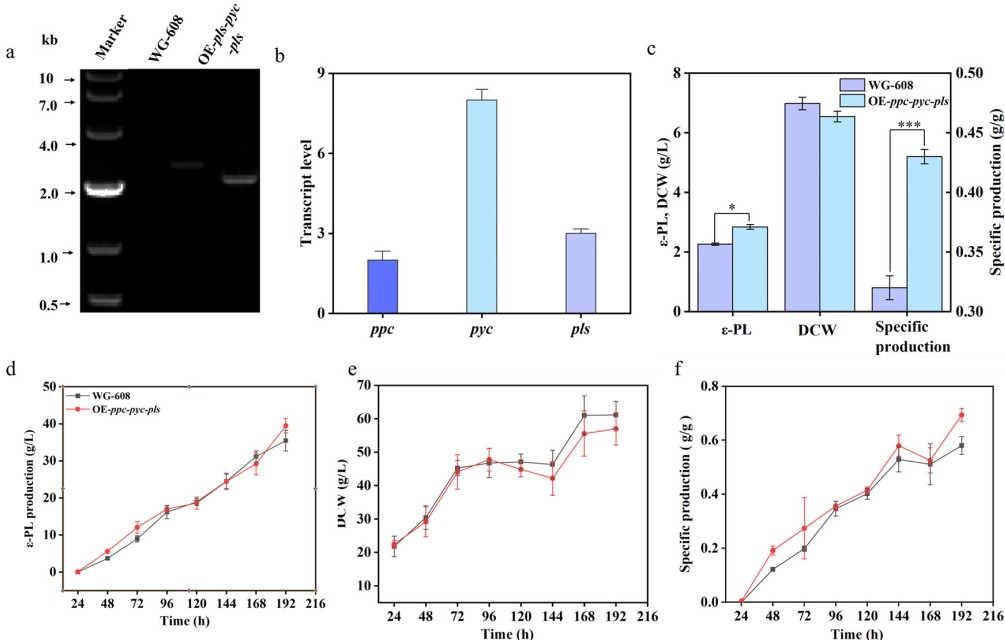

**Figure 5.** Construction and validation of OE-*ppc-pyc-pls*. (**a**) Genomic PCR. (**b**) RT-PCR. (**c**) Shake flask fermentation. (**d**–**f**) Fed-batch fermentation. (**d**) ε-PL production. (**e**) DCW. (**f**) Specific ε-PL production. The data are presented as averages, and the error bars represent standard deviations (*n* = 3). * $0.01 < p < 0.05$, *** $p < 0.001$.

#### 3.4.1. Shake-Flask Fermentation of OE-*ppc-pyc-pls* Strains

*S. albulus* WG-608 and OE-*ppc-pyc-pls* were subjected to shake-flask fermentation, as shown in Figure 5c. The ε-PL production of the *S. albulus* WG-608 was $2.3 \pm 0.07$ g/L, the DCW was $7.0 \pm 0.7$ g/L, and the specific ε-PL production was $0.3 \pm 0.01$ g/g. While the ε-PL production of OE-*ppc-pyc-pls* reached 2.8 g/L, which was 28.7% higher than *S. albulus* WG-608, the DCW was $6.5 \pm 0.5$ g/L, which was slightly lower than *S. albulus* WG-608. The specific ε-PL production reached $0.4 \pm 0.02$ g/g, which was 34.4% higher than *S. albulus* WG-608. From the shake-flask fermentation results, the co-expression of multiple genes could effectively improve the ε-PL production and the specific ε-PL production of *S. albulus* WG-608. However, similar to the results of single gene overexpression, the strain

showed some reduction in DCW. In the genetic manipulation of the strain, the strain was passed through too many generations, which led to a serious problem of decreasing the overall ε-PL production of the strain in shake-flask fermentation, which also indicated that co-expression of genes could effectively improve the ε-PL synthesis capacity of the strain.

3.4.2. Fed-Batch Fermentation of OE-*ppc-pyc-pls* Strains

Fed-batch fermentation is the dominant form of ε-PL production. To evaluate the enhancing effect of OE-*ppc-pyc-pls* on ε-PL production, fermentation of OE-*ppc-pyc-pls* was carried out under fed-batch fermentation conditions. As shown in Figure 5d–f, the ε-PL production of *S. albulus* WG-608 was 35.5 ± 1.3 g/L, and the specific ε-PL production was 0.6 ± 0.04 g/g. The ε-PL production of OE-*ppc-pyc-pls* was 39.5 ± 2.5 g/L, which was 11.4% higher than that of *S. albulus* WG-608, and the specific ε-PL production was 0.7 ± 0.02 g/g, which was 25.5% higher than *S. albulus* WG-608. We also found that the glucose consumption rates of OE-*ppc-pyc-pls* reached 7.2 g/h, which was slightly higher than *S. albulus* WG-608. This shows that the co-expression of multiple genes in *S. albulus* WG-608 increased the ε-PL production and specific ε-PL production during the fed-batch fermentation.

The cell growth of OE-*ppc-pyc-pls* was slightly reduced compared to *S. albulus* WG-608. At the same time, the glucose consumption rate increased to a certain extent compared with *S. albulus* WG-608. This may be because the insertion of the long gene puts a greater burden on the growth of the bacterium and the strain needs to consume more carbon sources to meet its own needs for energy and other substrates to maintain the normal growth of the strain. From the fermentation process (Supplementary Materials: Figure S2), OE-*ppc-pyc-pls* showed relatively low viability compared to *S. albulus* WG-608. This further suggests that the insertion of long gene fragments in *S. albulus* WG-608 had a detrimental effect on the growth of the strain.

## 4. Discussion

According to the metabolic flux distribution of OE-*ppc-pyc-pls* and *S. albulus* WG-608, it is easy to find that the flux of ε-PL synthesis in OE-*ppc-pyc-pls* was increased by 45.8% compared to *S. albulus* WG-608, as shown in Figure 6. The fluxes of the EMP pathway, anaplerotic reactions pathway, DAP pathway, and glutamate synthesis were also significantly increased, by 12.6%, 48.5%, 48.5%, and 50.7%, respectively. In contrast, the flux of the TCA pathway only slightly increased. However, the flux of the pentose phosphate pathway in OE-*ppc-pyc-pls* was unexpectedly decreased by 45.8%, presumably due to the enhancement of the anaplerotic reactions pathway, which led to the increase in carbon flux of the EMP pathway and then led to the significant decrease in carbon flux of the pentose phosphate pathway. Among them, the pentose phosphate pathway and TCA pathway can provide NADPH and ATP for ε-PL synthesis, glutamate can provide ammonium ion for ε-PL synthesis, while the anaplerotic reactions and DAP pathway flux can provide the main carbon skeleton of precursor L-lysine for ε-PL. From the changes in this metabolic flux, the co-expression of several genes did alter the primary carbon metabolic flux but did not achieve the expected effect of promoting the synthesis of ε-PL. This result confirms our speculation that insufficient ATP, NADPH, etc., probably caused the accelerated depletion of carbon sources during the 5 L fed-batch fermentation. We also saw some problems with this result, such as a significant decrease in the pentose phosphate pathway and a significant increase in the glutamate synthesis pathway, which provides us with ideas for our next study. Since the synthesis of L-lysine requires a large amount of ammonium ions and NADPH, and the synthesis of ε-PL requires a large supply of ATP, the next step can be considered to enhance the pentose phosphate pathway, glutamate synthesis, and energy supply based on OE-*ppc-pyc-pls*.

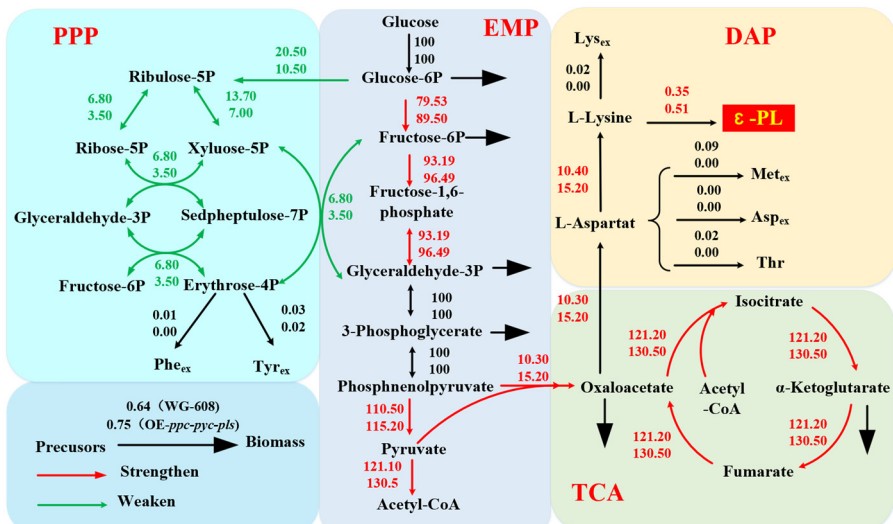

**Figure 6.** Metabolic flux distribution of OE-*ppc-pyc-pls* and *S. albulus* WG-608. Red arrows show the increased metabolic flux of OE-*ppc-pyc-pls*, the green arrow represents the decreased metabolic flux of OE-*ppc-pyc-pls*, the number above represents *S. albulus* WG-608, and the number below represents OE-*ppc-pyc-pls*. Red color indicates that the metabolic flux of OE-*ppc-pyc-pls* is stronger than WG-608; Green color indicates that the metabolic flux of WG-608 is stronger than OE-*ppc-pyc-pls*; Black indicates that there is no differences between the metabolic flux of OE-*ppc-pyc-pls* and WG-608.

## 5. Conclusions

In this study, metabolic flux analysis was employed to compare the differences between ε-PL high- and low-producing strains to identify potential nodes for metabolic engineering modification. Three effective key genes (*ppc*, *pyc*, and *pls*) were identified for the first time in ε-PL biosynthesis through overexpression experiments. Subsequently, we co-expressed *ppc*, *pyc*, and *pls* in *S. albulus* WG-608 to obtain strain OE-*ppc-pyc-pls*, and this engineered strain produced 39.5 g/L ε-PL in the fed-batch fermentation, which was an increase of 11.4% compared to *S. albulus* WG-608, and the specific ε-PL production was increased by 25.5%. However, we found that the DCW of OE-*ppc-pyc-pls* decreased significantly compared with *S. albulus* WG-608. Meanwhile, the glucose consumption rate increased compared with *S. albulus* WG-608, which may be due to the insertion of the long gene bringing a greater load on the growth of the strain. After analyzing the metabolic flux between OE-*ppc-pyc-pls* and *S. albulus* WG-608, it was found that OE-*ppc-pyc-pls* achieved the expected effect of altering the metabolic flux of ε-PL biosynthesis. Moreover, defects in the strain were also found, which paved the way for the next step of the work to a certain extent. This study not only provides a theoretical basis for the metabolic engineering of ε-PL-producing strains but also provides an effective approach for the metabolic engineering of other metabolites.

**Supplementary Materials:** The following supporting information can be downloaded at: https://www.mdpi.com/article/10.3390/fermentation10010065/s1, Table S1, Primers used in this study; Table S2, Chemical reaction rate equations in *S. albulus* WG-608; Table S3, Reaction rate equation of metabolic node; Table S4, Change in metabolite concentration and metabolic flux; Figure S1, Metabolic network of ε -polylysine synthesis; Figure S2, Fermentation process of *S. albulus* WG-608 and OE-*ppc-pyc-pls*. a: *S. albulus* WG-608. b: OE-*ppc-pyc-pls*. c: Comparison of fermentation parameters between *S. albulus* WG-608 and OE-*ppc-pyc-pls*.

**Author Contributions:** Data curation, C.Z. and L.W.; Writing—original draft, H.Y.; Writing—review & editing, H.Z. and D.Z.; Supervision, J.Z.; Funding acquisition, X.C. All authors have read and agreed to the published version of the manuscript.

**Funding:** This work was supported by the National Key R&D Program of China (2020YFA0907700); Key R&D Program of Jiangsu Province (BE2022703); and Evaluation Results of 2022 Open Subjects of Key Experiment of Ministry of Education in Industrial Biotechnology (KLIB-KF202204).

**Institutional Review Board Statement:** Not applicable.

**Informed Consent Statement:** Not applicable.

**Data Availability Statement:** Data are contained within the article and Supplementary Materials.

**Conflicts of Interest:** The authors declare no conflicts of interest.

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
