# Peer review of "Identification and Combinatorial Overexpression of Key Genes for Enhancing ε-Poly-L-lysine Biosynthesis in Streptomyces albulus"

_fermentation, doi:10.3390/fermentation10010065_

Round 1

Reviewer 1 Report

Comments and Suggestions for Authors

Overall, a sound ms. on the genetic engineering of Streptomyces to improve poly-lysing biosynthesis.

I do not have any comments on the science presented.

Comments on the Quality of English Language

Some minor comments to improve presentation follow.

lines 14-15. Change to "produced by the aerobic, filamentous bacterium Streptomyces....".

line 16  Change to "efficiency to reduce production....:.

line 17  Change to "few".

lines 23, 24, 241, 286, 287 and elsewhere. Reduce the significant figures of data; e.g., 11.4%, 25.4%, 5.0%....

line 36  Gram is a person's name, so it should be capitalized.

lines 46, 47, 397, 401, 402, 404, 405, 406. Add italics.

Table 1  Why is there a line in the table after "Overexpression of the vgb" line entry?

line 76  Omit "Judith".

line 83  Omit "firstly".

line 97  "was used as the".

line 119, 123, 223 and 505. Omit "the"; e.g., "To achieve overexpression of....".

line 123. "Separate sentences:  "China.  The DNA fragments obtained were....".

line 147. "with a working volume....".

line 151. Omit"then".

line 153. Omit "spontaneously".

line 154  Omit "(12.5%, v/v)", as this was established on line 150.

line 167. "sterile glucose".

lines 168 and 170. Omit "source".

line 175  "10 min and the precipitate".

line 177. Omit "centrifugal".

line 184. Should "Sangon" be capitalized?

lines 185, 344, 412 and 456. Omit the comma.

line 205. "B. subtilis".

line 221. "synthesis.  Therefore....".

line 230. "conducted in triplicate.".

lines 244. "significant differences in e-PL synthesis....".

lines 245, 282 and 493. "Should "notes" be "nodes"?

line 248. Omit "Our findings indicated that".

line 251. Omit "has".

line 259. Omit "independent experiments".

Fig. 3. Legend for 3 e and 3 f?  The appropriate information is given in the text, though.

lines 437-441. I am unsure of the meaning of the sentence starting with "And with the genetic....".

line 454. Omit "has".

line 477. Omit "highly significant".

line 482. "ideas for our next study.".

Reviewer 2 Report

Comments and Suggestions for Authors

This manuscript aims to enhance production of poly-L-lysine by metabolic flux optimization. Coupling of transcriptomics to aid and interpret metabolic flux data in conjunction with the production of poly-L-Lysine production is a major strength. The results do offer a realistic picture of metabolic flux for poly-L-lysine production, which can be valuable for further strain optimization through further pathway engineering or other means.

The manuscript is also full of errors, which need to be fixed.

While this manuscript reports the enhancement of PL production by 11.39% through targeted gene expression, the production yield is still almost half of the production achieved by S. albus R6 (the strain produced through mutagenesis and antibiotic screening). The discussion section should provide a possible explanation for such a high production or what can be done to enhance production (e.g. inactivation of certain flux-diverting genes?)

Page2 line 57: rewrite the sentence (….by metabolic engineering to remove>>>> through metabolic engineering by allowing for removal of feedback inhibition…)

Page 3, Line 97: ----is sed… (unclear what sed means?? Is it meant for “used”?); please put citation number at the end of the sentence.

Figure 1 c and d: PL production is exponentially growing until the last recorded time period (72 h). This is needs to be continued until it plateaus. (Major), error bars are missing.

Figure 5 d and f: the experiment should be conducted until PL production hits plateaus. The error bars are missing for all graphs.

Page 12 line 422: “---- compared to S. albus WG-608”. There is no data for S. albus WG-608 to compare in the picture.

Page 14, line 493: “… potential notes for metabolic…” (Notes does not make any sense here. Is it meant for node??)

Comments on the Quality of English Language

There are bunch of typos which need to be addressed.
